# VEGFR-2 conformational switch in response to ligand binding

Sarvenaz Sarabipour[1], Kurt Ballmer-Hofer[2], Kalina Hristova[1]*

[1]Department of Materials Science and Engineering, Johns Hopkins University, Baltimore, United States; [2]Laboratory of Biomolecular Research, Molecular Cell Biology, Paul Scherrer Institute, Villigen, Switzerland

**Abstract** VEGFR-2 is the primary regulator of angiogenesis, the development of new blood vessels from pre-existing ones. VEGFR-2 has been hypothesized to be monomeric in the absence of bound ligand, and to undergo dimerization and activation only upon ligand binding. Using quantitative FRET and biochemical analysis, we show that VEGFR-2 forms dimers also in the absence of ligand when expressed at physiological levels, and that these dimers are phosphorylated. Ligand binding leads to a change in the TM domain conformation, resulting in increased kinase domain phosphorylation. Inter-receptor contacts within the extracellular and TM domains are critical for the establishment of the unliganded dimer structure, and for the transition to the ligand-bound active conformation. We further show that the pathogenic C482R VEGFR-2 mutant, linked to infantile hemangioma, promotes ligand-independent signaling by mimicking the structure of the ligand-bound wild-type VEGFR-2 dimer.

*For correspondence: kh@jhu.edu

**Competing interests:** The authors declare that no competing interests exist.

## Introduction

Angiogenesis, the development of new blood vessels from pre-existing ones, plays a critical role in embryogenesis, organ development, and wound healing (*Qutub et al., 2009*; *Nessa et al., 2009*; *Adams et al., 2007*). In addition, angiogenesis is associated with diabetic and age-related macular degeneration and is required for the growth of solid tumors, which recruit blood vessels to ensure adequate supply with oxygen and nutrients (*Luo et al., 2008*; *Tsuzuki et al., 2001*). Angiogenesis is regulated by the ligands and receptors of the vascular endothelial growth factor (VEGF) signaling network (*Ferrara et al., 2003*; *Matsumoto et al., 2001*; *Koch et al., 2011*; *Olsson et al., 2006*; *Shibuya et al., 2006*). Of the three VEGF receptors, VEGFR-2 is the primary regulator of endothelial cell proliferation and migration (*Takahashi et al., 2005*; *Gerhardt et al., 2003*). VEGFR-2 is expressed in all human vascular endothelial cells and is often overexpressed in highly malignant solid tumors (*Smith et al., 2010*; *Yamagishi et al., 2013*). Aggressive cancerous phenotypes correlate with enhanced VEGFR-2 signaling (*Chatterjee et al., 2013*).

VEGFR-2 is a receptor tyrosine kinase (RTK) which transduces biochemical signals via lateral dimerization in the plasma membrane. Like most RTKs, VEGFR-2 is composed of an extracellular (EC) domain, a transmembrane (TM) domain, and an intracellular (IC) domain consisting of a kinase domain and sequences required for downstream signaling. The EC domain consists of seven immunoglobulin homology (Ig) domains, termed D1 (at the N-terminus) to D7 (closest to the membrane). VEGFR-2 binds to, and is activated by the ligands VEGF-A, VEGF-E, and a number of processed forms of VEGF-C and VEGF-D (*Olsson et al., 2006*; *Matsumoto et al., 2001*). Ligand binding to VEGFR-2 is mediated by Ig-domains 2 and 3 and the linker between D2 and D3 (*Brozzo et al., 2012*; *Leppanen et al., 2011*; *Kisko et al., 2011*; *Hyde et al., 2012*; *Ruch et al., 2007*).

All VEGF ligands form disulfide-linked anti-parallel homodimers (*Muller et al., 1997*). VEGF-A exists as different isoforms of various lengths, all containing the binding site for VEGFR-2, located

**eLife digest** New blood vessels form by growing out from existing vessels. A signaling molecule called VEGF is crucial for this process and binds to a receptor protein known as VEGFR-2. This binding activates signaling events within the cells that line the blood vessels to promote the growth of new vessels.

VEGFR-2 belongs to a family of proteins called receptor tyrosine kinases. The receptor has three main parts: one part extends out of the cell and binds to VEGF, another spans the cell's membrane, while the third part is found inside the cell. The current model of VEGFR-2 activation is that VEGF binds to individual VEGFR-2 receptor proteins on the membrane, and brings two of them close enough to form a complex called a dimer. The receptor dimer is activated and initiates signaling within the cell.

Sarabipour et al. tested this current model and revealed that VEGFR-2 could form a dimer even in the absence of VEGF. This receptor dimer formed through contacts between parts that are inside the cell as well as the regions embedded in the membrane. These VEGFR-2 dimers were active, but only to a low level. However, the activity of the receptor was increased in the presence of VEGF. Sarabipour et al. found that binding of VEGF promoted VEGFR-2 to change its three-dimensional shape, which made it more active. Further work suggested that VEGFR-2 dimers must first correctly assemble in the absence of VEGF so that active VEGFR-2 dimers can form at a later stage. Hence a VEGFR-2 dimer on its own is an important intermediate that is needed for full activation of the receptor.

A mutation in the gene that encodes VEGFR-2 causes a disorder called infantile hemangioma that leads to abnormal vessel formation in young children. Sarabipour et al. found that this mutation causes VEGFR-2 to change its shape even when no VEGF is present and thus makes the receptor overly active. Finally, tumors need to form new blood vessels to grow and survive. These new findings suggest that drugs that interact with VEGFR-2 dimers in the absence of VEGF may be able to prevent the receptor from becoming activated and could thus help cut the blood supply to tumors and treat cancers.

within the N-terminal 121 amino acids. All isoforms exhibit similar binding affinity to VEGFR-2 (*Mac Gabhann et al., 2010*; *Qutub et al., 2009*). The longer VEGF-A isoforms possess additional sequences which enable binding to co-receptors such as neuropilins and heparan sulfate proteoglycans (*Qutub et al., 2009*; *Vempati et al., 2014*).

At present, there is no universal consensus model of RTK-mediated transmembrane signaling. Instead, two models, which are not necessarily mutually exclusive, are discussed. The first is the 'diffusion-based' or 'canonical' model, in which RTKs exist as monomers in the absence of ligand because they do not have a sequence-specific propensity for lateral interactions. Ligand binding induces RTK dimerization, which brings the two catalytic domains into close proximity so they can cross-phosphorylate each other (*Fantl et al., 1993*). The second model is the 'pre-formed dimer model', in which the receptors can form dimers in the absence of ligand because they have a sequence-encoded propensity to interact laterally even in the absence of ligand, and thus they exist in a monomer-dimer equilibrium in the absence of ligand. Ligand binding promotes structural changes in the EC domain that trigger transmembrane signaling by reorienting the catalytic kinase domains (*Belov et al., 2012*; *Bocharov et al., 2008*; *2013*; *Sarabipour and Hristova, 2016*).

VEGFR-2 has been believed to follow the canonical model of ligand-induced dimerization and activation, with dimeric VEGF ligands binding to freely diffusing monomeric receptors and promoting receptor dimerization (*Kisko et al., 2011*; *Hyde et al., 2012*; *Ruch et al., 2007*; *Shibuya et al., 2006*; *Koch et al., 2011*). In the active dimers, homotypic receptor-receptor contacts in domains D4-7 were observed that appear critical for efficient phosphorylation of the VEGFR-2 receptors in the presence of ligand (*Yang et al., 2010*; *Hyde et al., 2012*; *Ruch et al., 2007*).

Ligand-independent dimer formation has been observed for several RTKs, such as EGFR, FGFR, and Trk (*Low-Nam et al., 2011*; *Maruyama et al., 2012*; *Mischel et al., 2002*; *Lin et al., 2012*; *Sarabipour and Hristova, 2016*). The dimerization propensity of VEGFR-2 in the absence of ligand,

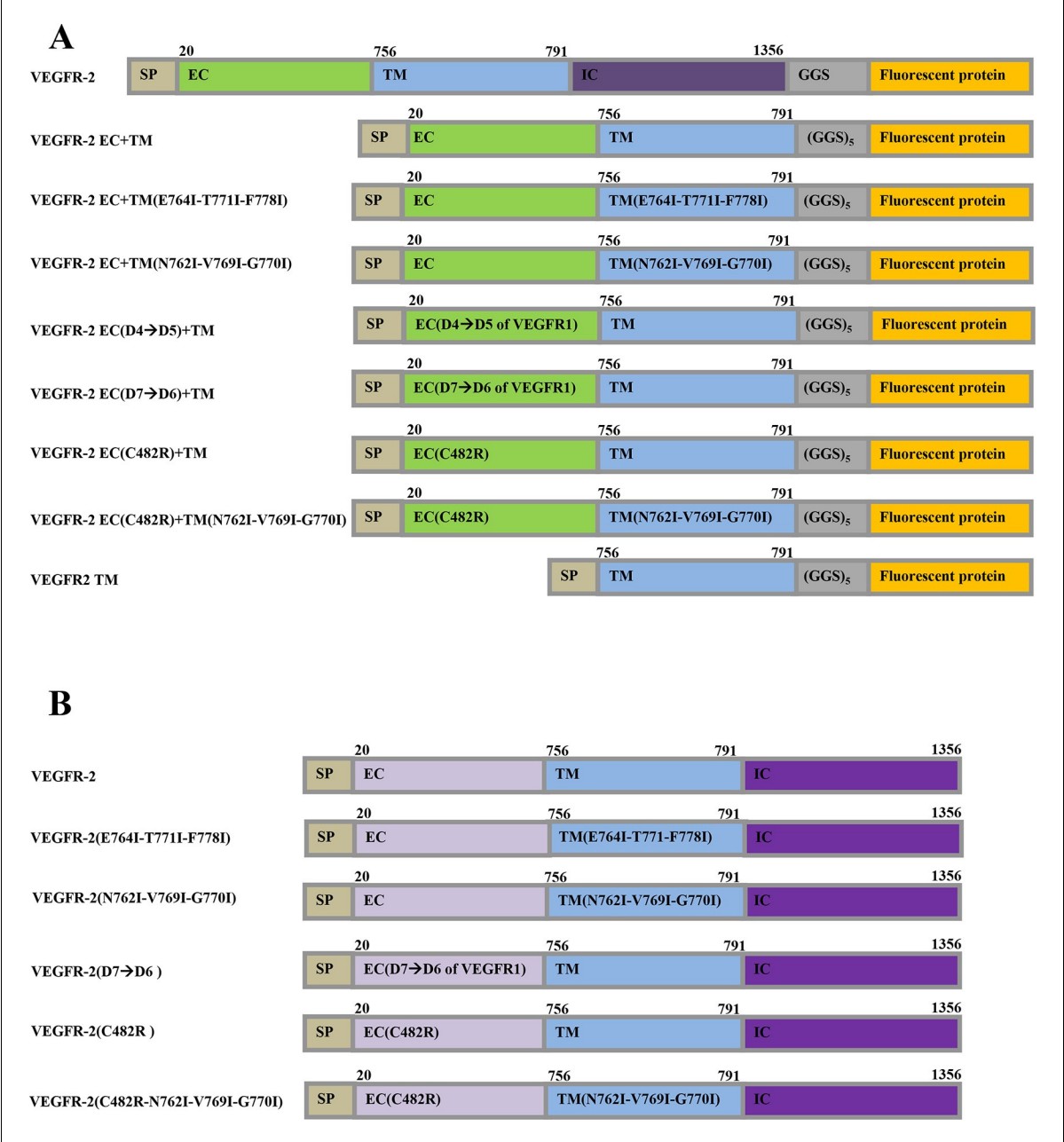

**Figure 1.** The plasmid constructs used in this study. (**A**) The constructs used in the FRET experiments. The full-length receptors had fluorescent proteins attached to their C-termini via a flexible GGS linker. The truncated receptors had the intracellular domain substituted with a fluorescent protein, which was attached to the TM domain via a longer flexible (GGS)$_5$ linker. SP: signal peptide, EC: extracellular domain, TM: transmembrane domain, IC: Intracellular domain of VEGFR-2. Fluorescent protein was either YFP or mCherry. Amino acid residue numbers are shown above the constructs. (**B**) The constructs used in the Western blotting experiments.

however, has not been measured to date. Here, we examined VEGFR-2 dimerization, as well as the mechanism of VEGFR-2 ligand-mediated activation. We used a quantitative FRET-based method and biochemical assays to study the structure, stability, and activation of VEGFR-2 dimers.

## Results

### Full-length VEGFR-2 dimerizes in the absence of ligand

We investigated whether full-length VEGFR-2 is capable of forming dimers in the absence of ligand using an established quantitative FRET method (*Chen et al., 2010*). The interactions between VEGFR-2 molecules, labeled with the fluorescent proteins YFP or mCherry (a FRET donor-acceptor pair), were characterized in plasma membrane-derived vesicles generated from transfected CHO cells in response to osmotic stress (*Del Piccolo et al., 2012*; *Sarabipour et al., 2015*).

While the plasma membrane topology of intact cells is complex (*Adler et al., 2010*; *Parmryd et al., 2013*), plasma-membrane-derived vesicles exhibit a simple membrane geometry, allowing us to determine the two-dimensional membrane receptor concentrations required for quantitative dimerization measurements (*Chen et al., 2010*). To determine the thermodynamic parameters describing VEGFR-2 dimerization, we varied the receptor concentration, and we measured the concentrations of the receptors and the FRET efficiencies in individual plasma membrane-derived vesicles (*Figure 2*). This method yields the dimerization constant $K$ and thus the dimerization free energy (or dimer stability) (*Chen et al., 2010*). In addition, the method allows to extract structural information from the 'Intrinsic FRET' value, which depends on the separation and the orientation of the fluorescent proteins in the dimer but not on the dimerization propensity (*Chen et al., 2010*; *Sarabipour and Hristova, 2015*). As shown previously, Intrinsic FRET values provide structural information by reporting changes in the spatial separation of the fluorescent proteins in receptor dimers (*Del Piccolo et al., 2015*; *Sarabipour and Hristova, 2015*) (see *Equation 7*).

The fluorescent proteins YFP and mCherry were attached to the C-terminus of VEGFR-2 via a flexible GGS linker (see *Figure 1*), to allow for their free rotation (*Evers et al., 2006*). CHO cells, which do not endogenously express measurable amounts of VEGFR-2 (*Figure 2—figure supplement 1*) were co-transfected with plasmids encoding VEGFR-2-YFP and VEGFR-2-mCherry. Following VEGFR-2 expression and trafficking to the plasma membrane, the cells were vesiculated by applying osmotic stress. Each vesicle was imaged in the donor, acceptor, and FRET channels (see *Figure 2—figure supplement 2*). The FRET efficiency, the donor concentration, and the acceptor concentration in each individual vesicle were determined as described in *Chen et al. (2010)* and in 'Materials and methods'. The measured FRET efficiency for VEGFR-2 is shown as a function of acceptor concentration (VEGFR-2-mCherry) in *Figure 2A* (red solid symbols). The solid black line shows the so-called 'bystander' or 'stochastic' FRET, which occurs due to random approach of donors and acceptors in the absence of specific interactions (*Wolber et al., 1979*). The magnitude of stochastic FRET is well understood and characterized, both theoretically and experimentally (*King et al., 2014*). The measured FRET efficiencies significantly exceed this stochastic FRET contribution, demonstrating the existence of specific interactions between the full-length VEGFR-2 molecules in the cell membrane.

VEGFR-2 dimerization in the absence of ligand is supported by VEGFR-2 cross-linking data (see *Figure 2—figure supplement 3*). Consistent with data in the literature, two monomeric bands were observed, corresponding to partially and fully glycosylated VEGFR-2 variants. We also observed weak bands at twice the VEGFR-2 molecular weight, corresponding to cross-linked dimers. We are aware that cross-linking efficiencies not only depend on dimerization, but also on the presence of suitable reactive groups in close proximity. Thus, the fact that we observed bands corresponding to dimers, albeit weak, provides support for our FRET results documenting VEGFR-2 dimerization in the absence of ligand.

The measured FRET in *Figure 2A* was corrected for the stochastic contribution, and the corrected FRET is plotted as a function of total receptor concentration in each vesicle in *Figure 2B* with the red solid symbols. The donor (VEGFR-2-YFP) concentration versus the acceptor (VEGFR-2-mCherry) concentration in each vesicle is shown in *Figure 2C* with the red solid symbols. The FRET data were then used to calculate the dimeric fraction as a function of total concentration as discussed in 'Materials and methods' and in previous publications (*Chen et al., 2010*). A model describing monomer-dimer equilibrium was fitted to the single vesicle data, yielding the optimal values for the dimerization constant $K$ and the Intrinsic FRET. The dissociation constant $K_{diss}=1/K$, the dimerization free energy $\triangle G$ (calculated using *Equation 10*) and the Intrinsic FRET value are shown in *Table 1*. The best fit-dimerization curve, plotted for the optimal parameters, is shown in *Figure 2D* in red, along with the experimentally measured binned dimeric fractions.

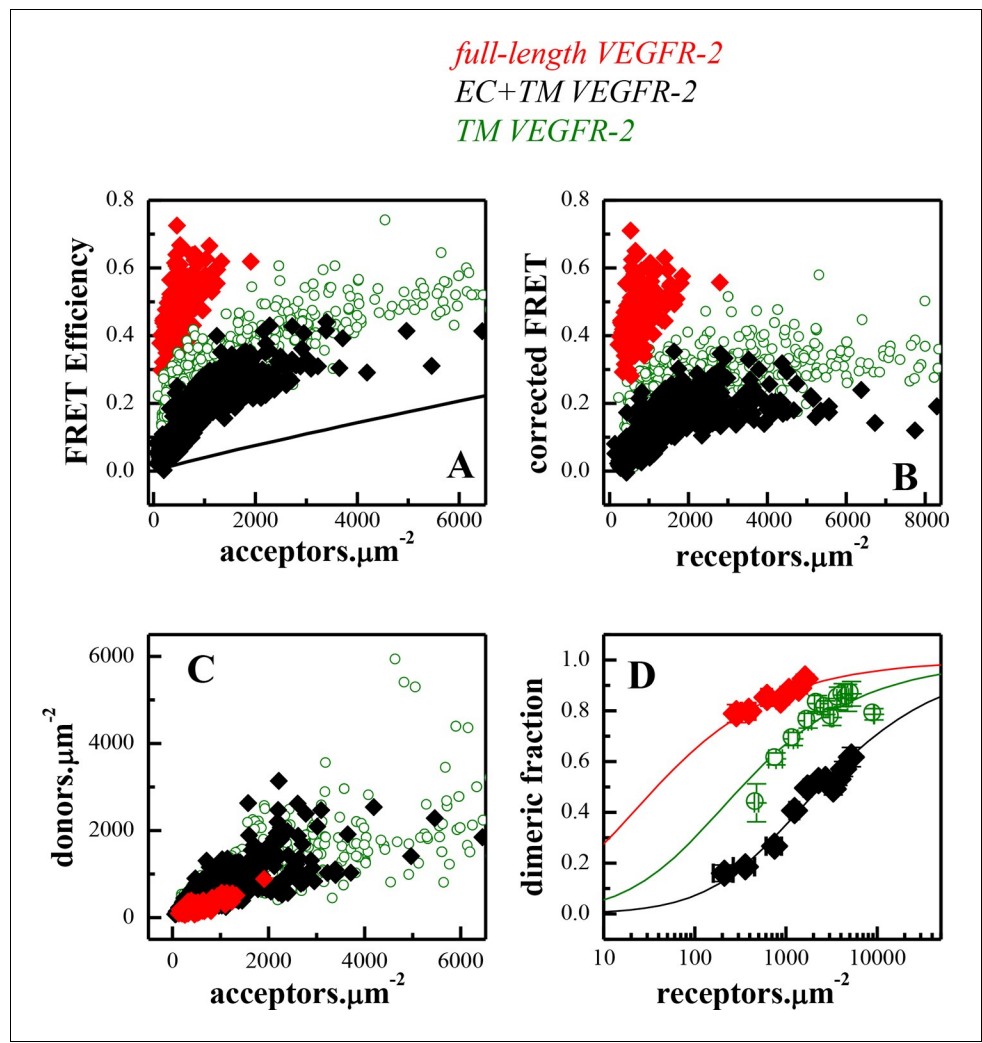

**Figure 2.** FRET measurements of VEGFR-2 dimerization in CHO plasma membranes. (A) FRET efficiencies as a function of acceptor concentration for full length VEGFR-2 (solid red diamonds), EC+TM VEGFR-2 (solid black diamonds), and TM VEGFR-2 (open olive circles). Two hundred to 500 individual vesicles were imaged in at least three independent experiments. Each data point here corresponds to a single vesicle. The solid line is the so-called 'stochastic' or 'proximity' FRET which arises when donors and acceptors approach each other within distances of 100 Å or so, in the absence of specific interactions (*King et al., 2014*). (B) FRET efficiencies corrected for stochastic FRET (*King et al., 2014*). (C) Donor concentration versus acceptor concentration in individual vesicles, for the three VEGFR-2 constructs. (D). Dimeric fractions versus total receptor concentrations, for the full-length VEGFR-2 (solid red diamonds), EC+TM VEGFR-2 (solid black diamonds), and the TM domains only (open olive circles). The measured dimeric fractions are binned and are shown with the symbols, along with the standard errors. The solid lines are the best fits of a monomer-dimer equilibrium model to the single-vesicle FRET data. Full-length VEGFR-2 has a significant propensity for dimerization in the absence of ligand. The contribution of the intracellular (IC) domain to dimerization is favorable, while the contribution of extracellular (EC) domain is inhibitory (see *Table 1*).

The following figure supplements are available for figure 2:

**Figure supplement 1.** (A) CHO cells do not express VEGFR-2 endogenously.

**Figure supplement 2.** A single vesicle imaged and analyzed in the FRET, acceptor, and donor channels.

**Figure supplement 3.** Cross-linking of CHO cells expressing full-length wild-type VEGFR-2.

**Table 1.** Dimerization free energies, $\Delta G$, and Intrinsic FRET efficiencies $\tilde{E}$, obtained from least-square parameter fits to the single-vesicle FRET data for the different VEGFR-2 constructs studied here. $d$ is the distance between the fluorescent proteins in the dimers, calculated from the the Intrinsic FRET values under the assumption of free fluorescent protein rotation. The uncertainties are 67% confidence intervals (standard errors) calculated from the fit in Matlab.

| VEGFR-2 variant | Dissociation constant, $K_{diss}$ (rec.$\mu$m$^{-2}$) | $\triangle$G (kcal.mol$^{-1}$) | I-FRET | d (Å) |
|---|---|---|---|---|
| full length | 35 (16 to 53) | -6.1 (-5.8 to -6.5) | 0.82 (0.78 to 0.85) | 41.3 (39.8 to 43) |
| EC+TM | 3200 (2712 to 3750) | -3.4 (-3.3 to -3.5) | 0.61 (0.55 to 0.68) | 49.3 (46.8 to 51.4) |
| TM | 310 (220 to 415) | -4.8 (-4.6 to -5.0) | 0.61 (0.59 to 0.65) | 49.3 (47.9 to 50.0) |
| EC+TM +VEGF-A$_{121}$ | 100% dimer | 100% dimer | 0.45 ± 0.02 | 55 ± 1 |
| EC+TM +VEGF-A$_{165}$a | 100% dimer | 100% dimer | 0.43 ± 0.02 | 56 ± 1 |
| EC+TM +VEGF-A$_{165}$b | 100% dimer | 100% dimer | 0.42 ± 0.02 | 56 ± 1 |
| EC+TM +VEGF-C | 100% dimer | 100% dimer | 0.45 ± 0.02 | 55 ± 1 |
| EC+TM +VEGF-D | 100% dimer | 100% dimer | 0.43 ± 0.02 | 56 ± 1 |
| EC+TM +VEGF-E | 100% dimer | 100% dimer | 0.44 ± 0.02 | 55 ± 1 |
| EC+TM(E764I-T771I-F778I) | 100% dimer | 100% dimer | 0.31 ± 0.02 | 61 ± 1 |
| EC+TM(E764I-T771I-F778I) + VEGF-A$_{121}$ | 100% dimer | 100% dimer | 0.38 ± 0.02 | 58 ± 1 |
| EC+TM(N762I-V769I-G770I) | 3100 (2090 to 3600) | -3.4 (-3.3 to -3.7) | 0.61 (0.58 to 0.65) | 49.3 (47.9 to 50.3) |
| EC+TM(N762I-V769I-G770I) + VEGF-A$_{121}$ | 100% dimer | 100% dimer | 0.24 ± 0.02 | 64 ± 1 |
| EC(D4→D5) +TM | 390 (271 to 480) | -4.7 (-4.5 to -4.9) | 0.82 (0.78 to 0.89) | 41.4 (37.5 to 43.0) |
| EC(D4→D5) + TM+VEGF-A$_{121}$ | 210 (130 to 350) | -5.0 (-4.7 to 5.3) | 0.83 (0.77 to 0.89) | 41.0 (37.5 to 43.4) |
| EC(C482R)+TM | 100% dimer | 100% dimer | 0.45 ± 0.02 | 55 ± 1 |
| EC(C482R)+TM+VEGF-A$_{121}$ | 100% dimer | 100% dimer | 0.47 ± 0.02 | 54 ± 1 |
| EC(C482R)+TM(N762I-V769I-G770I) | 100% dimer | 100% dimer | 0.31 ± 0.02 | 61 ± 1 |

The VEGFR-2 dimerization free energy, i.e. dimer stability, in the absence of ligand was -6.1 ± 0.4 kcal.mole$^{-1}$ (*Table 1*). To evaluate the significance of VEGFR-2 dimerization in the absence of ligand under physiological conditions, we note that (i) VEGFR-2 expression levels in endothelial cells are $10^4$ to $10^5$ copies per cell, corresponding to 10 to 100 receptors per square micron (*Napione et al., 2012*; *Lee et al., 2007*; *Imoukhuede et al., 2011*; *2012*), and that (ii) the measured dimerization curve in *Figure 2D* shows 30% to 60% VEGFR2 dimers in this expression range. Thus, VEGFR-2 has a strong dimerization propensity even in the absence of ligand.

## The extracellular domain inhibits VEGFR-2 dimerization while the intracellular domain promotes dimerization, in the absence of ligand

Next, the contributions of the three VEGFR-2 domains, the EC, the TM, and the IC domain, to the stability of dimers were evaluated in the absence of ligand. Experiments were performed with two truncated VEGFR-2 constructs, one lacking the IC domain (EC+TM construct) and one lacking both the EC and IC domains. The VEGFR-2 TM domain construct contained a (GGS)$_5$ flexible linker, and a fluorescent protein. This type of attachment of the fluorescent probe to the TM domains has previously been used successfully and has been demonstrated not to affect dimerization propensity (*Sarabipour and Hristova, 2015*). The EC+TM construct also included the VEGFR-2 EC domain. Both truncated constructs included the VEGFR-2 signal peptide which is cleaved during receptor processing. The FRET results for these two constructs are shown in *Figure 2* with the olive and black symbols and are summarized in *Table 1*.

The data given in *Figure 2* show that: (i) the isolated TM domain forms dimers (dimer stability $\triangle$G = -4.8 ± 0.2 kcal.mole$^{-1}$). (ii) the EC domain inhibits dimerization ($\triangle\triangle$G = +1.4 ± 0.3 kcal.mole$^{-1}$), such that the stability of the EC+TM construct dimer is reduced to -3.4 ± 0.2 kcal.mole$^{-1}$. This result is consistent with previous findings showing that the phosphorylation of VEGFR-2 is increased upon removal of the EC domain (*Manni et al., 2014a*). (iii) The IC domains stabilize the dimer by -

2.7 $\pm$ 0.3 kcal.mole$^{-1}$. Thus, we show that the EC domain inhibits VEGFR-2 dimerization in the absence of ligand, while the TM and IC domains enhance its dimerization propensity.

The FRET data further show that the TM and EC+TM constructs exhibit the same Intrinsic FRET value, 0.61, suggesting that the inclusion of the EC domain in these constructs does not change the separation between the fluorescent proteins in the dimer. Since the fluorescent proteins are attached directly to the TM domains via flexible linkers, this suggests that the addition of the EC domain does not have a measurable effect on the TM domain dimer.

## Ligand binding induces a structural change in the TM domain of the VEGFR-2 dimer

Our results thus far showed that a fraction of VEGFR-2 forms phosphorylated dimers in the absence of ligand, in a concentration dependent manner. Next, we investigated if ligand binding alters the structure of the VEGFR-2 dimer. We hypothesized that ligand-induced structural changes in the EC domain propagate to the TM domain, and we reasoned that such a structural change could manifest itself as a change in Intrinsic FRET for the EC+TM VEGFR-2 constructs, where the reporter fluorescent proteins are attached to the TM domains via $(GGS)_5$ flexible linkers.

The Intrinsic FRET for EC+TM of VEGFR-2 in the absence of ligand was 0.61 (*Table 1*). We performed FRET experiments in the presence of very high, saturating ligand concentrations, expected to yield 100% ligand-bound dimers. Under these conditions, the measured FRET signal would be independent of the receptor concentration and, instead, would depend solely on the donor-to-acceptor ratio and on the value of the Intrinsic FRET. Thus, the Intrinsic FRET can be measured directly in individual vesicles.

FRET data, measured in the presence of 3 $\mu$g.ml$^{-1}$ of VEGF, are shown in *Figure 3*. In *Figure 3A, B, and C*, we show single vesicle FRET results for the ligands VEGF-A$_{121}$ and VEGF-A$_{165}$a. Results for VEGF-A$_{165}$b, VEGF-C, VEGF-D, and VEGF-E were very similar. Consistent with our hypothesis, the corrected FRET efficiency did not vary with receptor concentration, indicative of constitutive dimerization. The Intrinsic FRET value was thus calculated for each individual vesicle, by dividing the corrected FRET efficiency by the acceptor fraction (see *Equation 6*). *Figure 3D* shows histograms of the Intrinsic FRET values for single vesicles in the presence of the six ligands. The histograms are well described by Gaussian distributions. The center of the Gaussians, $\sim$0.45, corresponding to a separation between the fluorescent proteins by $\sim$56 Å, was the same for all six ligands. Since the fluorescent proteins were attached to the TM domains via flexible linkers, we conclude that the TM domain configuration is similar in the presence of the six ligands. On the other hand, the binding of any of the ligands significantly changed Intrinsic FRET for the EC+TM construct from 0.61 to 0.45, thus increasing the separation of the fluorescent proteins by 7 $\pm$ 1 Å, from 49 $\pm$ 2 Å to 56 $\pm$ 1 Å. This demonstrates a ligand-induced structural change in the VEGFR-2 dimer, which promotes an increase in the spatial separation of the TM domain C-termini.

We next compared the phosphorylation of full-length VEGFR-2 in the absence of ligand, and in the presence of 3 $\mu$g.ml$^{-1}$ VEGF-A$_{121}$, VEGF-A$_{165}$a, VEGF-A$_{165}$b, VEGF-C, VEGF-E, and VEGF-D. *Figure 3F/G* shows that receptor phosphorylation is the same for all six ligands, and increases as much as 10-fold compared to the phosphorylation of VEGFR-2 in the absence of ligand.

The FRET and activation results thus clearly demonstrate a correlation between EC+TM VEGFR-2 dimer structure and receptor activation: the lower the Intrinsic FRET, that is, the larger the separation between the TM domain C-termini, the higher the receptor phosphorylation. Furthermore, the identical Intrinsic FRET values observed in the presence of the six ligands at saturating concentrations correlate with identical VEGFR-2 phosphorylation. Thus, the TM domain structure exerts control over VEGFR-2 phosphorylation and activation.

## VEGFR-2 dimers are stabilized by distinct, highly specific interactions between the TM domains

The structure of the isolated wild-type VEGFR-2 TM domain dimer was recently solved in lipid bicelles (*Manni et al., 2014b*). To evaluate whether this structure agrees with the FRET data, we created two sets of EC+TM mutant constructs tagged with YFP and mCherry. First, amino acids E764, T771, and F778 were mutated to Ile to create the E764I-T771I-F778I mutant. These three amino acids stabilize the TM helix contacts in the NMR-derived TM dimer model (*Manni et al., 2014b*).

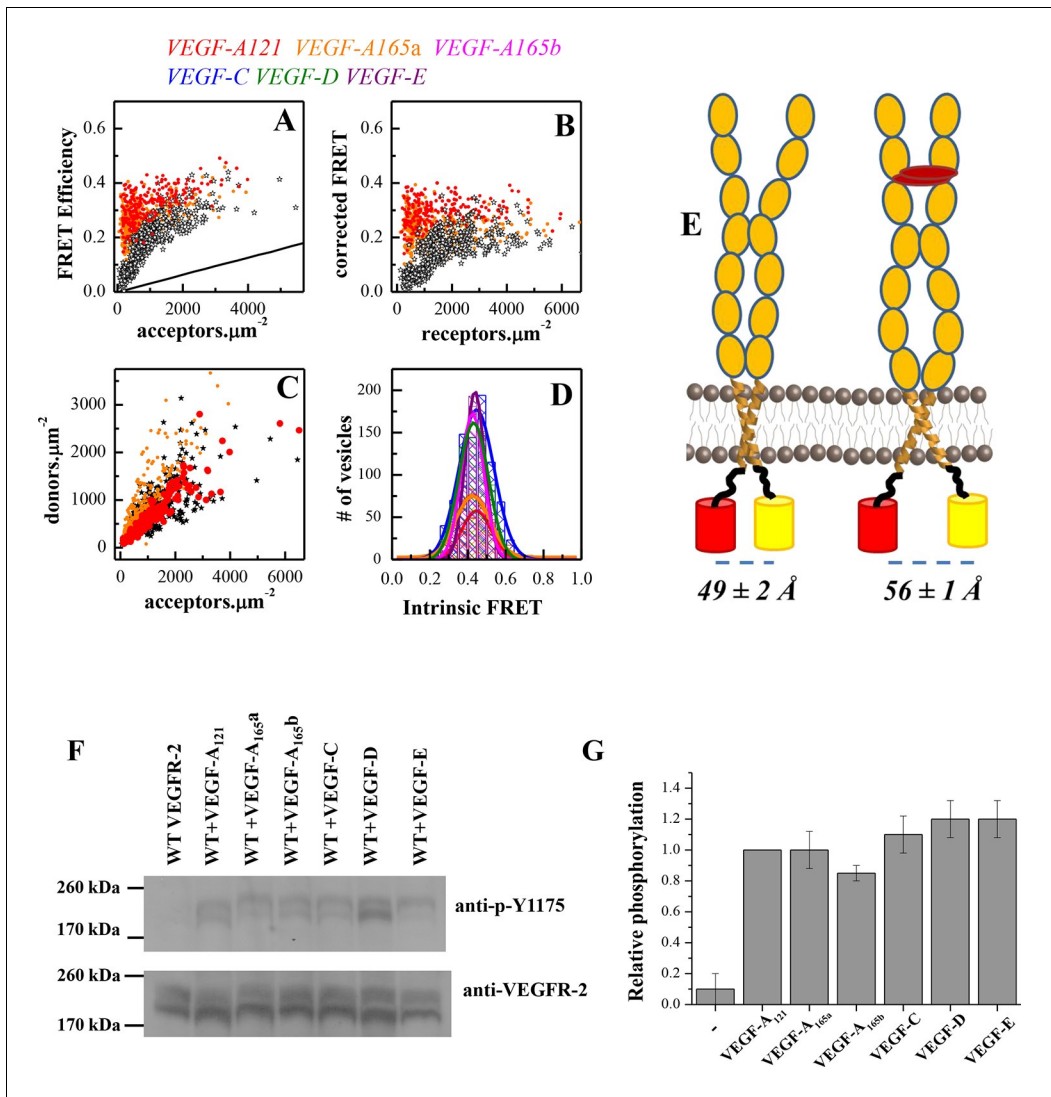

**Figure 3.** A conformational change in the TM domain dimer upon ligand binding increases VEGFR-2 phosphorylation. (A) FRET efficiency measured as a function of acceptor concentration for EC+TM VEGFR-2, in the absence of ligand and in the presence of 3 $\mu$g.ml$^{-1}$ VEGF-A$_{121}$ and VEGF-A$_{165}$a. Two hundred to 500 individual vesicles were imaged in at least three independent experiments. Each data point corresponds to a single vesicle. The stochastic FRET contribution (*King et al., 2014*) is shown as a solid line. Black stars: FRET data without ligand. (B) FRET efficiencies for individual vesicles, corrected for the stochastic FRET contribution. There is no dependence on receptor concentration, indicative of constitutive dimerization. (C) Donor concentrations versus acceptor concentrations. (D) Intrinsic FRET values, measured for VEGFR-2 EC+TM in the presence of 3 $\mu$g.ml$^{-1}$ of VEGF-A$_{121}$, VEGF-A$_{165}$a, VEGF-A$_{165}$b, VEGF-C, VEGF-E, and VEGF-D. (E) Graphical representation of the ligand-induced changes in distance between fluorescent proteins, and the inferred changes in TM domain structures. (F) The six VEGF ligands increase VEGFR-2 phosphorylation, to the same extent. A representative Western Blot comparing the phosphorylation of Tyr 1175 in the absence of ligand, and in the presence of six VEGF ligands, at concentrations of 3 $\mu$g.ml$^{-1}$. Only the top bands, corresponding to mature fully glycosylated receptors, are considered here. Phosphorylation is significantly increased upon ligand addition, with the increase being as high as 10 times. (G) Quantification of Western blot results for the fully glycosylated receptors (top bands), in the presence of the six ligands. VEGFR-2 phosphorylation in the presence of the six ligands is very similar.

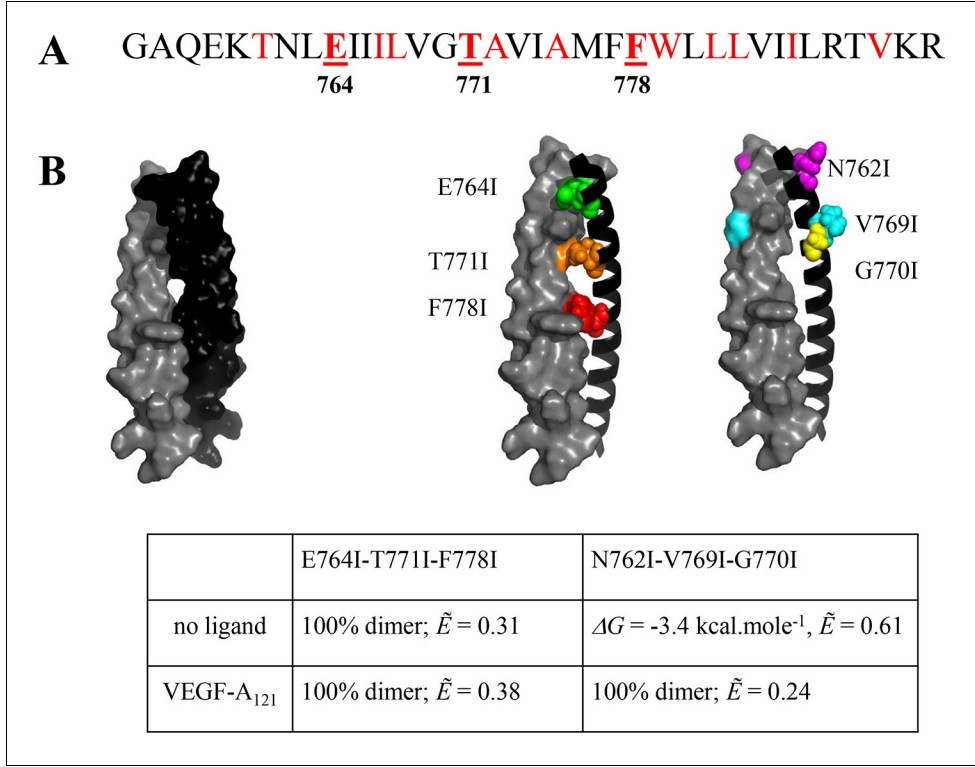

**Figure 4.** The published NMR structure of the isolated VEGFR-2 TM domain (*Manni et al., 2014b*) is consistent with the unliganded TM dimer structure observed in the FRET experiments. (**A**) Amino acid sequence of wild-type VEGFR-2 TM domain. The amino acids that mediate helix-helix contacts in the NMR dimer structure (*Manni et al., 2014b*) are shown in red. (**B**) The NMR structure, with two sets of amino acids highlighted. Left: a space-fill model of the wild-type VEGFR-2 TM dimer, based on NMR experiments. Center: E764, T771, and F778 help mediate helix-helix contacts in the NMR structure (*Manni et al., 2014b*) (also shown bold and underlined in (**A**)). Right: N762, V769, and G770 face away from the dimer interface, into the membrane. Two sets of mutations: a E764I-T771I-F778I set and a N762I-V769I-G770I set, were engineered. Table: Results of FRET experiments for the two sets of mutants (see also *Table 1*). The dimerization of the unliganded EC+TM construct was affected by the E764I-T771I-F778I but not by the N762I-V769I-G770I set of mutations, suggesting that the unliganded VEGFR-2 TM dimer is stabilized by contacts involving E764, T771, and/or F778, as in the NMR structure. At least one of the amino acids N762, V769, and G770 forms direct intermolecular contacts in the TM domain in the ligand-bound state.

The following figure supplements are available for figure 4:

**Figure supplement 1.** FRET data, reporting on the effects of the E764I-T771I-F778I and N762I-V769I-G770I sets of mutations, engineered in the EC+TM VEGFR-2 plasmids.

**Figure supplement 2.** All mutations in the TM domain decrease VEGFR-2 phosphorylation.

Separately, a N762I-V769I-G770I mutant was generated from the EC+TM-YFP and EC+TM-mCherry constructs. These residues do not contribute to the TM domain dimer interface in the NMR-derived model as they face the lipid side instead.

The FRET results for the two EC+TM mutants are summarized in *Figure 4* (raw FRET data are shown in *Figure 4—figure supplement 1*). Dimerization in the absence of ligand was enhanced and the Intrinsic FRET was altered for the E764I-T771I-F778I mutant, as compared to wild-type. Thus, the E764I-T771I-F778I set of mutations affected the structure of the TM dimer in the absence of ligand. On the other hand, the N762I-V769I-G770I mutations had no effect on the FRET efficiencies measured for these constructs in the absence of ligand, suggesting that the published NMR structure of the isolated wild-type TM domain dimer represents the TM dimer configuration of the unliganded EC+TM constructs embedded in the plasma membrane.

The N762I-V769I-G770I mutant significantly affected the Intrinsic FRET of the EC+TM dimer in the VEGF-A$_{121}$ bound state. The ligand-induced phosphorylation of the full-length N762I-V769I-G770I mutant was lower as compared to wild-type VEGFR-2 (*Figure 4—figure supplement 2*). These findings suggest that (i) at least one of the amino acids N762, V769, or G770 mediate helix-helix contacts in the ligand-bound state, and (ii) the conformation of the ligand-bound EC+TM dimer differs from that observed in the wild-type VEGFR-2 TM domain NMR structure. Our mutagenesis data thus show that different contacts between the TM domains stabilize the unliganded and the ligand-bound VEGFR-2 dimers. The data also provide direct experimental support for the concept that the TM domain dimer switches between two different configurations when transitioning from the unliganded to the ligand-bound state, as suggested previously (*Manni et al., 2014b*).

## Ig-like domains D4 and D7 regulate VEGFR-2 dimerization in the absence of ligand

The current model of VEGFR-2 activation postulates that the ligand-bound EC domain dimer is stabilized by ligand binding to domains D2 and D3. In addition, homotypic contacts between Ig domains D4-D7 have been observed in the isolated ligand-bound dimeric EC domains (*Ruch et al., 2007*). These contacts were proposed to promote and stabilize a specific conformation of the two receptor monomers in ligand-bound VEGFR-2 dimers. Mutagenesis demonstrated that these contacts are indispensable for activation of the intracellular kinase domains (*Hyde et al., 2012*).

Here, we investigated how D4 and D7 affect dimerization of the EC +TM construct in the ligand-free and ligand-bound states. We examined two EC+TM VEGFR-2 mutants in which D4 or D7 were substituted with unrelated domains from VEGFR-1, D4(VEGFR-2)→D5(VEGFR-1) and D7(VEGFR-2)→D6(VEGFR-1). FRET data for these mutant constructs are shown in *Figure 5*. The FRET efficiencies of the mutants were much higher than for the wild-type EC+TM construct indicating a change in dimer stability and/or conformation.

A monomer-dimer equilibrium model was successfully fitted to the FRET data for the EC+TM VEGFR-2 D4(VEGFR-2)→D5(VEGFR-1) construct (*Figure 5—figure supplement 1*). Dimerization of the D4 mutant was stronger than for the wild type, and the mutant dimer exhibited much higher Intrinsic FRET. Thus, the substitution of the D4 domain caused a structural change in the unliganded VEGFR-2 EC domain favoring non-productive dimer formation in the absence of ligand. On the other hand, a monomer-dimer model could not be fitted to the D7 mutant data. This is indicative of the formation of higher order oligomers (*King et al., 2016*). Thus, the substitution of both D4 and D7 significantly altered the association of the VEGFR-2 molecules in the absence of ligand.

The FRET efficiencies measured for the two mutants remained unchanged upon treatment with saturating amounts (3 $\mu$g.mL$^{-1}$) of VEGF-A$_{121}$ (*Figure 5A*), suggesting that neither of the mutants respond to ligand. To investigate whether these mutants were capable of binding ligand, we incubated the vesicles expressing these mutants with fluorescently labeled ligand (VEGF-A$_{121}$-AF594). In the case of ligand binding, vesicle fluorescence would be observed both in the green channel, because of the presence of wild-type or mutant VEGFR-2-YFP, and in the red channel, because of VEGF-A$_{121}$-AF594 binding to the receptor. No fluorescence would be observed in the red channel, however, if ligand does not bind. The results in *Figure 5B* showed that ligand binds to the D7 mutant and to the wild-type receptor, but not to the D4 mutant.

We thus conclude that substitutions of D4 and D7 domains rendered the VEGFR-2 unresponsive to ligand, but in different ways. The D4 mutant dimers were structurally different from wild-type and were incapable of binding ligand. The D7 mutants formed oligomers in the absence of ligand, and ligand binding did not reverse oligomerization and did not promote transition to the correctly assembled ligand-bound dimeric state of the receptor. The latter conclusion is supported by results in *Figure 5C*, showing that ligand binding to the D7 mutant does NOT increase the phosphorylation of the D7 mutant.

Taken together, results in the absence and presence of ligand demonstrate that domains D4 and D7 are essential for establishing the properly oriented unliganded, as well as ligand-bound, VEGFR-2 dimers. We therefore propose that the unliganded state is a critically important intermediate in VEGFR-2 activation.

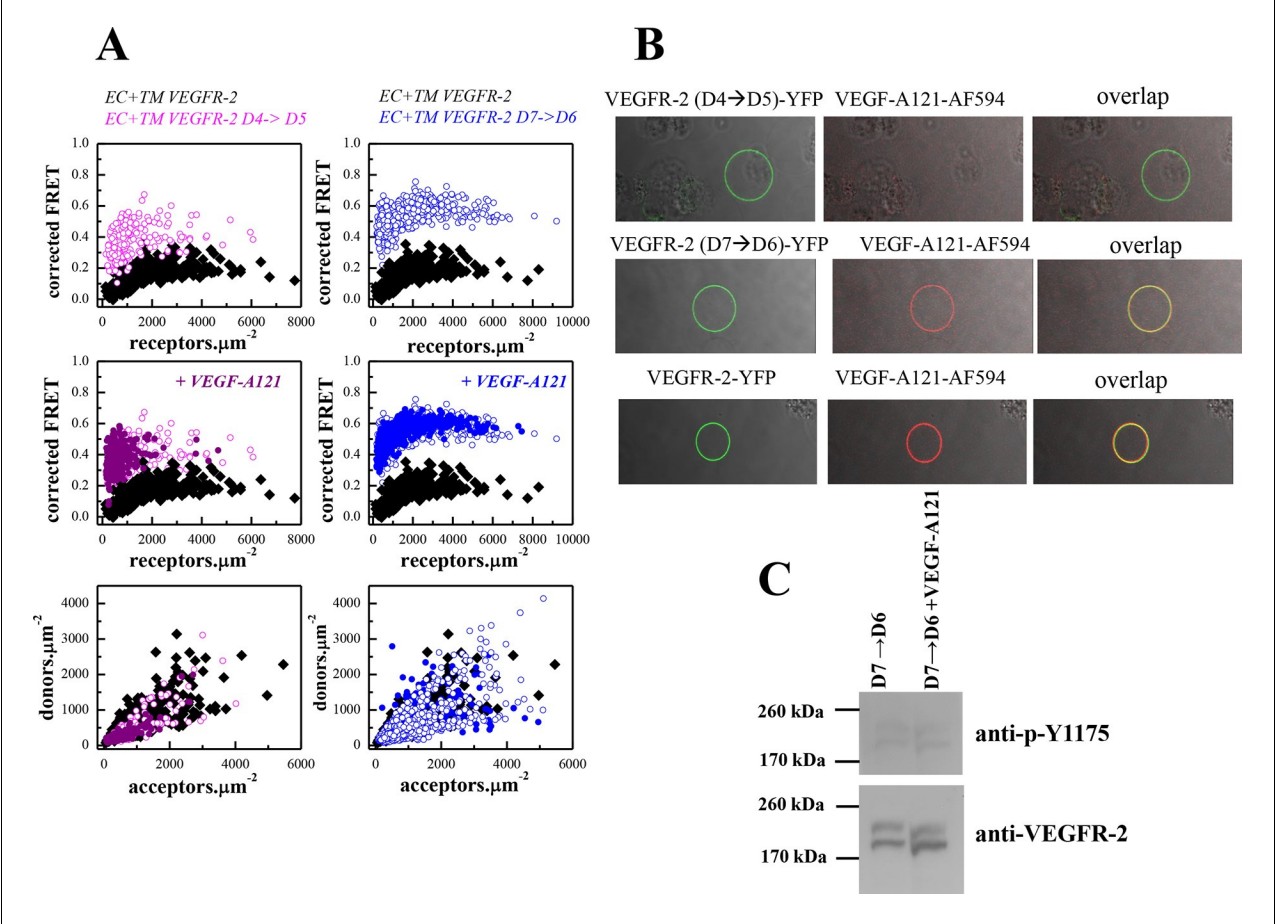

**Figure 5.** The D4 and D7 mutants do not respond to ligand. (**A**) FRET measurements for the D4 and D7 mutants, in the absence of ligand and in the presence of VEGF-A$_{121}$. The two mutants exhibit much higher FRET efficiencies, compared to the wild-type. The ligand, at 3 $\mu$g.ml$^{-1}$, has no effect on receptor dimerization. (**B**) VEGF-A$_{121}$ does not bind to the D4→D5 mutant, when it binds to the wild-type and the D7→D6 mutant. Here, a single vesicle containing YFP-tagged mutant receptors (two top rows) or wild-type receptors (bottom row) is shown after incubation with 3 $\mu$g.ml$^{-1}$ AlexaFluor 594-labeled VEGF-A (VEGF-A$_{121}$-AF594) (SibTech Inc., CT). Note the differences in the middle column, showing the fluorescence of the bound ligand. (**C**) Western blots comparing the phosphorylation of the full-length D7→D6 mutant in the absence and presence of VEGF-A$_{121}$. The top band corresponds to the mature fully glycosylated form of VEGFR-2. The ligand does not increase the phosphorylation.

The following figure supplement is available for figure 5:

**Figure supplement 1.** Dimerization curves for the wild-type VEGFR-2 in the absence of ligand, the D4→D5 mutant in the absence of ligand, and the D4→D5 mutant in the presence of VEGF-A$_{121}$.

## The C482R EC mutation induces ligand-independent constitutive receptor dimerization and activation

The C482R mutation in D5 of VEGFR-2 has been implicated in infantile hemangioma, characterized by disorganized angiogenesis in infants (*Boye et al., 2009*). The FRET data for the mutant are shown in *Figure 6 A–D*. The fraction of dimeric C482R EC+TM receptor was similar in the presence and absence of ligand and was not dependent on receptor concentration. The Intrinsic FRET values measured in the presence and absence of ligand were the same, and similar to those measured for ligand-bound wild-type VEGFR-2 dimers. Thus, it appears that this mutation locked the VEGFR-2 dimer in the ligand-bound conformation in the absence of ligand. Consistent with these structural observations, the phosphorylation in the presence and absence of ligand was the same (*Figure 6E*), despite the fact that this mutant was capable of binding fluorescent ligand (*Figure 6F*). The data in *Figure 6* therefore demonstrate the formation of constitutive C482R mutant dimers both in the

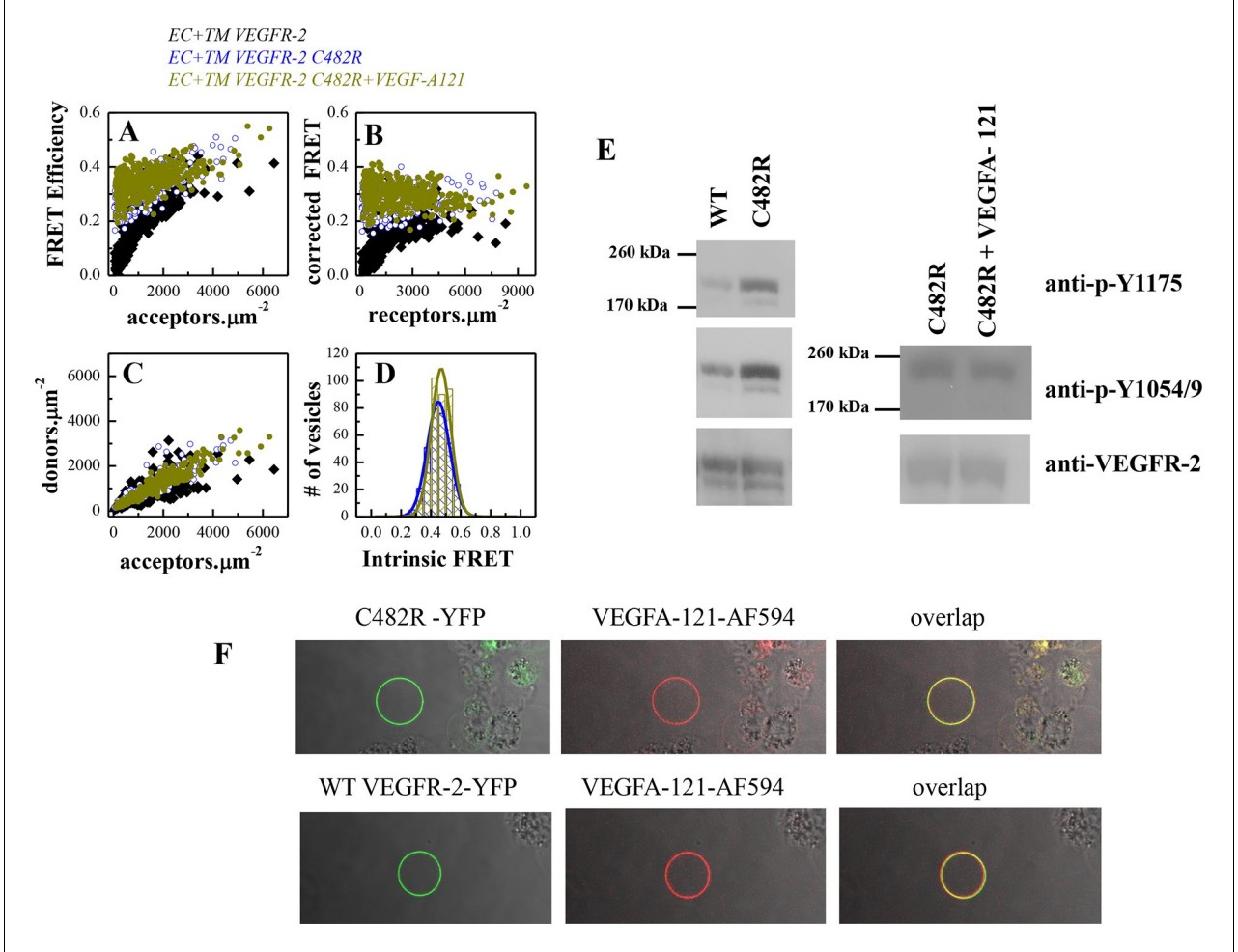

**Figure 6.** The C482R mutation mimics the effect of bound ligand by promoting a structural change in the EC+TM VEGFR-2 dimer. (A) FRET efficiencies determined in individual plasma-membrane-derived vesicles. (B) Corrected FRET as a function of receptor concentration. The corrected FRET for the mutant does not depend on receptor concentration, demonstrating that the mutant is a constitutive dimer in the presence and absence of ligand. (C) Measured donor versus acceptor concentrations in each vesicle. (D) Histograms of measured FRET efficiencies for the mutant, yielding Intrinsic FRET values of ∼0.42. (E) Western blots showing an increase in phosphorylation due to the C482R mutation, in the absence of ligand, and no further increase upon ligand treatment. The top bands correspond to the mature fully glycosylated form of VEGFR-2. (F) Confocal images of VEGF-A$_{121}$-AF594 binding to EC+TM VEGFR-2 C482R in CHO membrane vesicles, demonstrating that the C482R mutant is capable of ligand binding.

absence and the presence of ligand. They also reveal a lack of structural transition upon ligand binding. Importantly, the phosphorylation of the mutant was higher than the phosphorylation of the wild-type in the absence of ligand (*Figure 6E*), demonstrating that this mutation causes pathological ligand-independent constitutive VEGFR-2 activation.

To confirm that the unliganded C482R dimer structure is similar to the ligand-bound wild-type VEGFR-2 dimer structure, we engineered the N762I-V769I-G770I mutations in the C482 mutant construct. The FRET data for the mutant EC+TM, shown in *Figure 7A and B*, demonstrate that the TM domain mutations altered the Intrinsic FRET for the C482 mutant, suggesting that at least one of these amino acids mediate helix-helix contacts in the C482 mutant, reminiscent of the case of the ligand-bound wild-type dimer. Consistent with this result, the TM domain mutations decreased the phosphorylation of the full-length C482R mutant receptor, as observed for the ligand-bound wild-type VEGFR-2 dimer (*Figure 7 C*).

We next investigated whether intermolecular disulfide bonds between possible unpaired cysteine residues in the full-length and EC+TM C482R mutant receptors promote constitutive dimerization of the receptor. We analyzed the dimerization state of the receptors under reducing and oxidizing

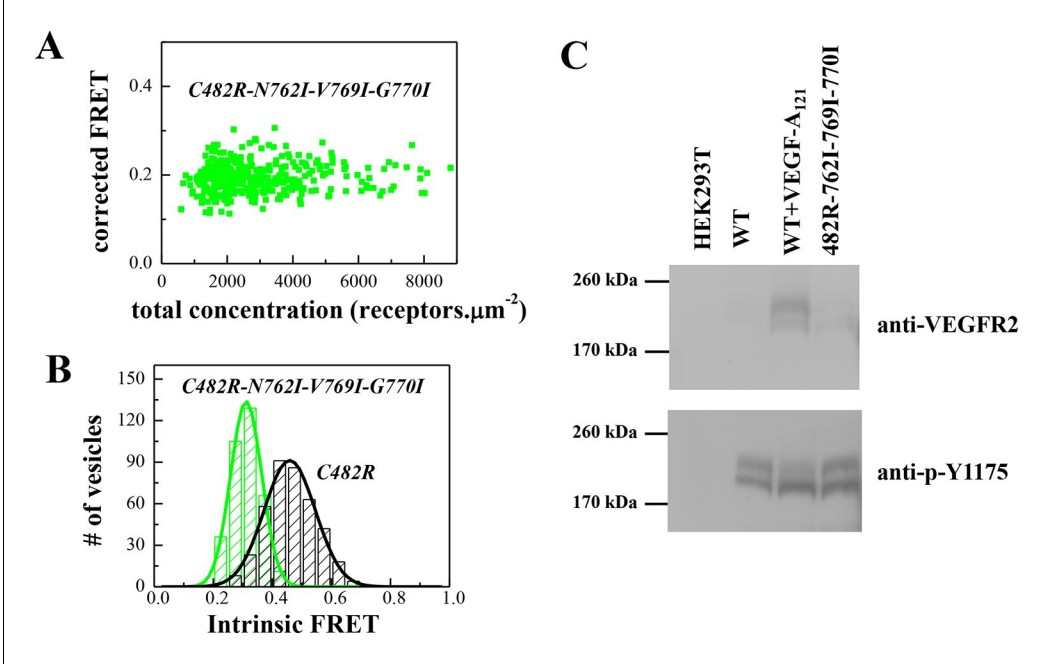

**Figure 7.** The N762I-V769I-G770I set of mutations in the TM domain of the C482R mutant receptor alters dimer structure and activity, as in the case of the ligand-bound wild-type (see **Figure 4**). (**A**) Corrected FRET efficiencies for the EC+TM C482R-N762I-V769I-G770I mutant as a function of receptor concentration, indicative of constitutive dimers. (**B**) Histograms of Intrinsic FRET measured for the EC+TM C482R mutant and the EC+TM C482R-N762I-V769I-G770I mutant. The TM domain mutations alter Intrinsic FRET and thus the structure of the C482R dimer. (**C**) The N762I-V769I-G770I set of mutations obliterate the activity of the C482R mutant, as in the case of the ligand-bound wild type.

The following figure supplement is available for figure 7:

**Figure supplement 1.** Left: Western blots under reducing conditions, for full length C482R VEGFR-2 and for the C482R EC+TM-YFP VEGFR-2 construct.

conditions using Western blotting. The results, shown in *Figure 7—figure supplement 1*, demonstrate that receptors were not disulfide-linked both in the presence and absence of reducing agent. Thus, the C482 mutation did not lead to direct intermolecular disulfide bond formation, which suggests that the structural effects of the C482R mutation are more complex. Perhaps, this mutation alters the protein fold. In addition, this mutation may affect interactions with other proteins such as co-receptors.

## Discussion

### VEGFR-2 forms dimers in the plasma membrane in the absence and presence of ligand

VEGFR-2 has been believed to exist in a monomeric form in the absence of ligand and has been hypothesized to follow the canonical model of ligand-induced dimerization and activation (*Yang et al., 2010*; *Hyde et al., 2012*; *Koch et al., 2011*; *Matsumoto et al., 2001*; *Olsson et al., 2006*). Using a quantitative FRET-based methodology, specifically designed to probe receptor interactions in the plasma membrane, we demonstrated that full-length VEGFR-2 forms dimers in the absence of ligand. A substantial fraction of the receptors is dimeric for receptor expression levels reported for endothelial cells (10 to 100 receptors per square micron). Furthermore, these unliganded dimeric receptors are phosphorylated, albeit to a low degree. The basal phosphorylation occurs on Y1054/1059 and Y1175 (*Figure 6E*), as well as on Y1214 (*Lamalice et al., 2006*).

Consistent with these observations, phosphorylation of downstream kinases has been reported in endothelial cells in the absence of ligand (*Oubaha et al., 2009*; *Labrecque et al., 2003*).

Recent work has demonstrated that other RTKs, such as EGFR and the FGF receptors (*Chung et al., 2010*; *Low-Nam et al., 2011*; *Sarabipour and Hristova, 2016*; *Lin et al., 2012*; *Comps-Agrar et al., 2015*) also form unliganded dimers. A new model of RTK activation, which includes unliganded dimers of various stabilities as intermediates, has been proposed (*Sarabipour and Hristova, 2016*). The behavior of VEGFR-2 observed here is consistent with this new model.

Through mutagenesis, domain deletions and domain substitutions, our study provides insights into the interactions that stabilize VEGFR-2 dimers in the plasma membrane in the absence of ligand. First, there are sequence-specific contacts between the TM domains in the dimers. In particular, at least one of the amino acids E764, T771, or F778 participates in helix-helix contacts, as the substitution of these three amino acids with Ile changed the structure and the stability of the EC+TM VEGFR-2 dimers in the absence of ligand. Second, these dimers are further stabilized by contacts between the IC domains, since the removal of the IC domains decreased dimer stability by 2.7 kcal.mole$^{-1}$. Therefore, specific interactions between the IC domains regulate receptor dimerization and ensure control of receptor activity in the absence of ligand. Thus, our observations that VEGFR-2 phosphorylation can take place in the absence of ligand, along with the FRET results, point to a significant role of the IC domains in dimer stabilization. Third, we discovered a new role for domains D4 and D7 in unliganded VEGFR-2 dimerization, by showing that they stabilize VEGFR-2 dimers in the absence of ligand. Indeed, the substitution of Ig-domains D4 or D7 significantly altered both dimerization and Intrinsic FRET in the absence of ligand. Fourth, we showed that the wild-type VEGFR-2 EC domain, as a whole, inhibits receptor dimerization, as inferred from the fact that deletion of the entire EC domain increased the stability of the dimers by 1.4 kcal.mole$^{-1}$. Overall, this work documents a complex role for the EC domain in VEGFR-2 dimer stabilization and establishes the existence of specific dimer stabilizing contacts in the EC, the TM, and the IC domains of VEGFR-2.

## Ligand binding promotes reorientation of the receptors in the dimer

To monitor ligand-induced conformational changes in the TM domain, we performed experiments with truncated VEGFR-2 constructs carrying fluorescent proteins attached via flexible (GGS)$_5$ linkers to the C-termini of the TM domains. The Intrinsic FRET decreased from 0.61 to 0.42, corresponding to an increase in separation between the fluorescent proteins in these dimers from 49 to 56 Å, upon ligand binding. This reflects an increase in the spatial separation between the TM domain C-termini by 7 Å in response to ligand binding.

Through mutagenesis analysis, we gained information about the structural changes in the TM domain of the VEGFGR-2 dimers that occur in response to ligand binding (see *Figure 8*). The configuration of the TM domain of the ligand-free EC+TM VEGFR-2 dimer was consistent with the published NMR spectroscopy-derived structural model of isolated wild-type VEGFR-2 TM domain dimers in lipid bicelles (*Manni et al., 2014b*). Ligand binding induces a rotation of the TM helices, leading to the engagement of at least one of the amino acids N762, V769, G770 in the helix-helix interface. These residues were chosen for mutagenesis because they face away from the TM helix interface in the NMR-derived model for the wild-type (*Manni et al., 2014b*). Substitution of these residues by Ile had no effect on dimerization in the absence of ligand, but had a significant effect in the presence of ligand.

The TM domains link the ligand-binding EC domains to the IC domains and are thus expected to play a role in the activation of the intracellular kinase in response to ligand binding. Previous NMR studies of in vitro produced RTK TM peptides, inserted into lipid bicelles or detergent micelles, have suggested the existence of two sets of mutually exclusive helix-helix contacts (*Bocharov et al., 2008*; *Li et al., 2010*; *Fleishman et al., 2002*). These observations have been interpreted as evidence that RTK TM domains can form at least two structurally and functionally different dimers. Yet, there have been no direct experimental data demonstrating a structural change in the VEGFR-2 TM domain dimer configuration in the plasma membrane, in response to ligand binding, so far. Here, we directly observed such a structural change using our quantitative FRET methodology which allowed us to monitor the response of the receptor dimer to ligand binding from the cytoplasmic

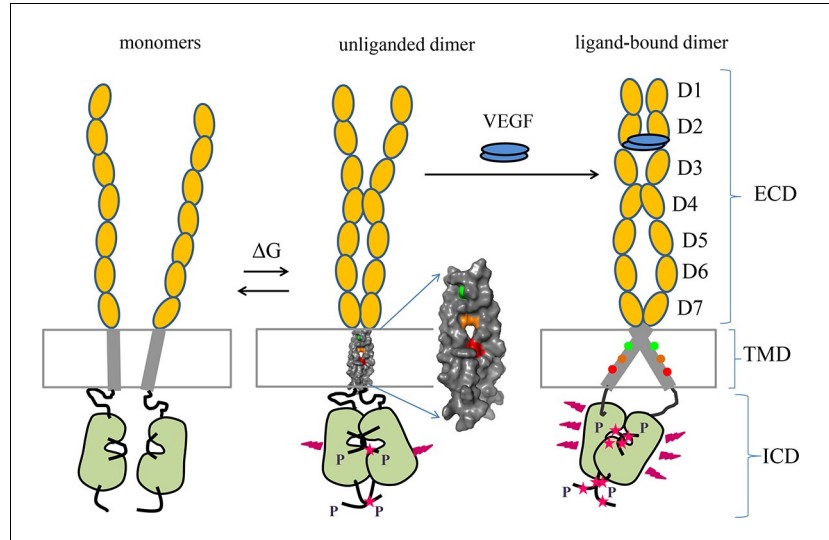

**Figure 8.** Proposed model for VEGFR-2 activation. VEGFR-2 pre-dimerizes in the absence of ligand. Under physiological conditions, corresponding to 10 to 100 VEGFR-2 molcules per square micron, 30 to 60% of the receptors are dimeric. The dimers are stabilized by homotypic contacts in D4-7 in the EC domain, TM interactions that involve amino acids E764, T771, and/or F778, and contacts between the intracellular domains. The EC domain, as a whole, inhibits dimerization. The ligand-free dimers are phosphorylated, to a low degree. (**B**) Ligand binding induces a rotation in the TM helices and the formation of a different TM dimer configuration, such that amino acids E764, T771, and F778 now face the lipid membrane, away from the TM dimer interface. These structural changes increase the separation between the C-termini of the TM helices, promoting an increase in receptor phosphorylation and activation. Contacts between D4 and D7 persist in the ligand-bound state.

side of the membrane. FRET is extremely sensitive to the distance between the fluorescent proteins, which we measure with 1 Å precision.

The observed changes in FRET are biologically significant, as the increase in separation between the C-termini of the TM domains correlated with an increase in receptor phosphorylation and thus with receptor activity. Our results clearly demonstrate that ligand binding increases VEGFR-2 phosphorylation, which correlates with a conformational change in the TM domain dimer structure. A similar mechanism of activation, involving an increase in TM helix C-termini separation, has been proposed for EGFR (*Bocharov et al., 2008*; *Arkhipov et al., 2013*; *Endres et al., 2013*). For the FGF receptors, however, the ligand-induced changes involve a decrease in the separation between the TM domain C-termini (*Sarabipour and Hristova, 2016*). Decrease in the separation of the TM domains upon ligand binding has also been observed for the constitutively dimeric IGF-1R (*Kavran et al., 2014*). Thus, it appears that ligand-induced structural changes are a general feature of RTK activation, but the exact nature of the structural change varies for the different receptors.

We compared the structural effects of several ligands known to bind to VEGFR-2, namely three different VEGF-A isoforms (VEGF-A$_{121}$, VEGF-A$_{165}$a, and VEGF-A$_{165}$b), as well as VEGF-C, VEGF-D, and VEGF-E. Our experiments were conducted at very high, saturating ligand concentrations of 3 $\mu$g.ml$^{-1}$, when ligand concentrations exceeded both the receptor concentrations and the ligand binding coefficients by at least a factor of 100. The FRET efficiency did not depend on receptor concentration, suggesting that all receptors were in the ligand-bound dimeric state. We measured the same Intrinsic FRET for EC+TM VEGFR-2 in the presence of all six ligands. Furthermore, these ligands gave rise to the same VEGFR-2 activities as determined by Western blotting with phospho-specific antibodies. We thus propose that the six ligands induced the same structural changes in the receptor, resulting in a similar VEGFR-2 phosphorylation increase. The VEGFR-2 case is thus quite different from the case of the FGFR dimers, which exhibit different structural and biological responses to the ligands fgf1 and fgf2 (*Sarabipour and Hristova, 2016*).

The VEGF ligands are known to have distinct affinities for VEGFR-2 (*Brozzo et al., 2012*; *Mac Gabhann et al., 2010*; *Qutub et al., 2009*). Thus, for VEGFR-2 it is reasonable to assume that

binding affinity may determine the specific signal output by VEGFR-2 in response to different ligands. Alternatively, different intracellular trafficking of the ligand-bound receptors may dictate signaling specificity (*Clegg et al., 2015*; *Ballmer-Hofer et al., 2011*).

## Effect of mutations on transmembrane signaling

The mutations and substitutions used here to probe VEGFR-2 dimer stability, structure, and activation, promoted significant changes in VEGFR-2 structure and function. For instance, the D4 substitution stabilized the EC+TM VEGFR-2 dimer, altered its structure, and abolished ligand binding. The D7 substitution, on the other hand, caused receptor oligomerization that rendered the receptor non-responsive to ligand. The E764I-T771I-F778I TM domain mutations stabilized the VEGFR-2 dimer in the absence of ligand and changed its structure as reflected by Intrinsic FRET, but did not increase receptor phosphorylation. The N762I-V769I-G770I mutations altered the structure of the ligand-bound VEGFR-2 dimer and reduced receptor phosphorylation. Quite remarkably, none of the engineered mutations increased VEGFR-2 phosphorylation, suggesting that VEGFR-2 is surprisingly resistant to aberrant activation.

The only mutation that caused an increase in phosphorylation was the pathogenic C482R mutation, identified in infant hemangiomas (*Boye et al., 2009*). Our studies therefore demonstrated that the C482R mutation is a gain-of-function mutation. To the best of our knowledge, this is the first report of a pathogenic activating mutation in VEGFR-2 and the first mechanistic study of such a mutation.

Our results demonstrate that the C482R mutant formed constitutive dimers even in the absence of ligand. Western blots performed under reducing and non-reducing conditions revealed that there are no disulfide-linked C482R dimers. Instead, quantitative FRET experiments revealed that the constitutive C482R dimers adopt the ligand-bound wild-type dimer conformation, both in the presence and absence of ligand. Thus, the C482R mutation increased VEGFR-2 activation by mimicking the structural effect of the ligand. Similar behavior has been observed for the pathogenic A391E, R248C, and S249C FGFR3 mutants, linked to skeletal and cranial growth disorders (*Sarabipour and Hristova, 2016*; *Del Piccolo et al., 2015*). Thus, this mechanism of over-activation may apply to RTKs from diverse families.

## Materials and methods

### Plasmids

The plasmids encoding for full length wild-type human VEGFR-2, full length C482R mutant, and full-length VEGFR-2 containing TM mutations, used in the Western Blot experiments, were constructed in the pBE vector. The full-length D7(VEGFR-2)→D6(VEGFR-1) mutant was in the pcDNA5/FRT vector. The full-length D4(VEGFR-2)→D5(VEGFR-1) mutant was in the pcDNA3 vector. All truncated VEGFR-2 plasmid constructs were engineered in the pcDNA 3.1(+) vector. All the plasmids contained the N-terminal VEGFR2 signal sequence, which is cleaved off during processing. All primers were purchased from Invitrogen (Calsbad, CA). The pRSETB-mCherry and pRSETB-YFP plasmids were received from Dr. M. Betenbaugh (Johns Hopkins University, Baltimore, MD) and Dr. R. Tsien (University of California, San Diego), respectively. Both fluorescent proteins are monomeric and do not drive interactions in the membrane (*King et al., 2014*)

YFP and mCherry were fused to the C-terminal end of all VEGFR2 constructs. To generate the VEGFR-2-(GGS)-mCherry and VEGFR-2-(GGS)-YFP plasmids, a MluI restriction site was created in the multiple cloning site of the pBE vector, which contained the full length human VEGFR-2 insert. The multiple cloning site was located right after the 3' end of the VEGFR-2 sequence. The cDNAs encoding YFP and mCherry were then amplified using Polymerase Chain Reaction (PCR) and double digested with MluI and XhoI. The double digested YFP and mCherry cDNAs were then ligated with the digested pBE-VEGFR cDNA.

The sequence encoding for the EC and TM domains of VEGFR-2 were amplified by PCR, double digested using HindIII and EcoRI restriction enzymes, and inserted into a pcDNA3.1(+) vector containing the flexible 15 amino acids $(GGS)_5$ linker and YFP or mCherry, producing the VEGFR-2 EC+TM-$(GGS)_5$-YFP and VEGFR-2 EC+TM-$(GGS)_5$-mCherry plasmids.

The C482R mutant plasmid constructs: VEGFR-2(C482R), VEGFR2 EC(C482R)+TM-(GGS)$_5$-YFP, and VEGFR-2 EC(C482R)+TM-(GGS)$_5$-mCherry were created using the QuikChange II XL Site-Directed Mutagenesis Kit (Stratagene, CA) from the wild-type constructs.

The full-length D4(VEGFR-2)→D5(VEGFR-1) construct was generated by replacing Ig domain 4 of VEGFR-2 with Ig domain 5 of VEGFR-1. The full-length D7(VEGFR-2)→D6(VEGFR-1) construct was created by replacing domain 7 of VEGFR-2 with domain 6 of VEGFR-1 (detailed in [*Hyde et al., 2012*]).

To construct the VEGFR-2 EC(D4→D5)+TM-(GGS)$_5$-YFP and VEGFR-2 EC(D4→D5)+TM-(GGS)$_5$-mCherry plasmids, cDNA encoding EC(D4→D5)+TM was amplified from the full-length VEGFR-2 (D4→D5) plasmid construct. This cDNA was double digested with Hind III and EcoRI. The digested cDNA was then ligated with pcDNA3.1(+)-(GGS)$_5$-YFP or pcDNA3.1(+)-(GGS)$_5$-mCherry, doubly digested with Hind III and EcoRI.

To construct the VEGFR-2 EC(D7→D6)+TM-(GGS)$_5$-YFP and VEGFR-2 EC(D7→D6)+TM-(GGS)$_5$-mCherry plasmids, cDNA encoding EC(D7→D6)+TM was amplified from the VEGFR-2(D7→D6) plasmid construct. This cDNA was double digested with Hind III and EcoRI. Finally, the digested cDNA was ligated with pcDNA3.1(+)-(GGS)$_5$-YFP or pcDNA3.1(+)-(GGS)$_5$-mCherry, doubly digested with Hind III and EcoRI.

To generate the VEGFR-2 TM-(GGS)$_5$-YFP and VEGFR-2 TM-(GGS)$_5$-mCherry plasmids, the sequence encoding the TM domain of VEGFR-2 (GAQEKTNLEIIILVGTAVIAMFFWLLLVIILRTVKR) was amplified using PCR. This cDNA was then double digested with HindIII and EcoRI and ligated with digested pcDNA3.1(+)-(GGS)$_5$-YFP or pcDNA3.1(+)-(GGS)$_5$-mCherry.

The N762I-V769I-G770I, E764I-T771I-F778I and C482R-N762I-V769I-G770I full length and truncated mutant VEGFR-2 constructs: VEGFR-2 (N762I-V769I-G770I), VEGFR-2 (E764I-T771I-F778I), VEGFR-2(C482R-N762I-V769I-G770I), VEGFR-2 EC + TM(N762I-V769I-G770I)-(GGS)$_5$-YFP, VEGFR-2 EC+TM(N762I-V769I-G770I)-(GGS)$_5$-mCherry, VEGFR-2 EC+TM(E764I-T771I-F778I)-(GGS)$_5$-YFP, VEGFR-2 EC +TM(E764I-T771I-F778I)-(GGS)$_5$-mCherry, VEGFR-2 EC(C482R) +TM(N762I-V769I-G770I)-(GGS)$_5$-YFP, and VEGFR-2 EC(C482R)+TM(N762I-V769I-G770I)-(GGS)$_5$-mCherry plasmids were created using the QuikChange II XL Site-Directed Mutagenesis Kit (Stratagene, CA) from the wild-type constructs.

## Cell culture and transfection for QI-FRET experiments

Chinese Hamster Ovary cell (CHO) cells were cultures at 37˚C with 5% CO$_2$ for 24 hr. Transfection was carried out using either Fugene HD or Lipofectamine, following the manufacturer's protocol. Cells were cotransfected with 3–7 ug of DNA encoding VEGFR2-YFP and VEGFR2-mCherry constructs. Cells were vesiculated 24 hr post transfection.

## Production of mammalian plasma membrane vesicles

Vesiculation was performed using a chloride salt vesiculation buffer (*Del Piccolo et al., 2012*) consisting of of 200 mM NaCl, 5 mM KCl, 0.5 mM MgSO$_4$, 0.75 mM CaCl$_2$, 100 mM bicine and protease inhibitor cocktail (Complete mini EDTA-free tabs, Roche Applied Science) adjusted to pH of 8.5. CHO cells were rinsed twice with 30% PBS (pH 7.4), and incubated with 1 mL of chloride salt vesiculation buffer overnight at $37 deg$C. Vesicles were produced after 12 hr, and transferred into four-well Nunc Lab-Tek II chambered coverslips for imaging.

The plasma membrane-derived vesicles produced with this method have been characterized previously (*Sarabipour et al., 2015*). They contain other membrane proteins and all the major lipid species found in the plasma membranes of mammalian cells, but lack cytoplasmic content.

## Treatment of plasma-membrane-derived vesicles with VEGF

One ml of vesicle solution was collected in a well of a 4 chamber slide and treated with 3000 ng.mL$^{-1}$ of either human VEGF-A$_{121}$ (#8908, Cell Signaling Technologies), human VEGF-A$_{165a}$ (#8065, Cell Signaling Technologies), human VEGF-C (#100-20C, PeproTech), human VEGF-D (#100-20D, PeproTech), or human VEGF-A$_{165b}$ and VEGF-E, purified as described in *Scheidegger et al. (1999)*. After 1 hr incubation with ligands at room temperature, the vesicles were imaged.

## QI-FRET data analysis: methodology and protocol

The QI-FRET method has been described in detail previously (*Chen et al., 2010*; *Del Piccolo et al., 2015*; *Sarabipour and Hristova, 2015*). Vesicles were imaged using a Nikon Eclipse confocal laser scanning microscope using a 60× water immersion objective. All the images were collected and stored at a 512 × 512 resolution. Three different scans were performed for each vesicle: (1) excitation at 488 nm, with a 500–530 nm emission filter (donor scan); (2) excitation at 488 nm, with a 565–615 nm emission filter (FRET scan); and (3) excitation at 543 nm, with a 650-nm longpass filter (acceptor scan).

Solutions of YFP and mCherry, which served as concentration calibration controls, were purified as described (*Sarabipour et al., 2014*), concentrated in vesiculation buffer, and imaged in the donor, acceptor and FRET channels. Using these solution standards, we determined the calibration constants for the donor and the acceptor, $i_D$ and $i_A$, and the bleed-through coefficients for the donor and the acceptor, $\beta_D$ and $\beta_A$ as previously described (*Li et al., 2008*). Vesicles loaded with a soluble linked YFP-mCherry protein were also imaged in the three channels to obtain the gauge factor $G_F$ (*Li et al., 2008*). The acceptor concentrations in each vesicle, $C_A$, was calculated according to:

$$C_A = \frac{I_A}{i_A} \qquad (1)$$

where $I_A$ is the intensity captured in the acceptor channel. The sensitized emission of the acceptor in each vesicle was determined as:

$$I_{SEN} = I_{FRET} - \beta_A I_A - \beta_D I_D \qquad (2)$$

Next, we calculate:

$$I_{D,corr} = I_D + G_F I_{SEN} \qquad (3)$$

$$C_D = \frac{I_{D,corr}}{i_D} \qquad (4)$$

where $I_{D,corr}$ is the reconstructed donor intensity in the absence of the acceptor, and $C_D$ is the donor concentration. The FRET efficiency, $E$, was calculated according to:

$$E = 1 - \frac{I_D}{I_{D,corr}} \qquad (5)$$

The FRET efficiency was corrected for the so-called proximity FRET, which occurs when donors and acceptors approach each other by chance within distances of 100 Å or so. This correction has been discussed in detail previously (*King et al., 2014*), and is required because the fluorescent proteins are confined to the two-dimensional vesicles.

The dimeric fraction is calculated from the corrected FRET, $E_D$, and the acceptor fraction, $x_A$, according to:

$$f_D = \frac{E_D}{x_A \tilde{E}} \qquad (6)$$

Here $\tilde{E}$ is the 'Intrinsic FRET', the FRET efficiency in a dimer containing a donor and an acceptor. This is a purely structural parameter, which depends only on the separation and the orientation of the two fluorescent proteins in the dimer, $d$, but not on the dimerization propensity.

$$\tilde{E} = \frac{1}{1 + \left(d/R_0\right)^6} \qquad (7)$$

Here $d$ is the distance between the acceptor and the donor in the dimer, and $R_o$ is the Förster radius of the FRET pair (*Chen et al., 2010*). For YFP and mCherry, $R_o$ is 53.1 Å.

Based on the law of mass action, the dimeric fraction can be written as a function of the total receptor concentration, $T$, and the dimerization constant $K$ according to *Equation 8*:

$$f_D = \frac{1}{T}\left(T - \frac{1}{4K}(\sqrt{1 + 8TK} - 1)\right) \qquad (8)$$

Substituting *Equation 8* into *6*, we obtain:

$$\frac{E_D}{x_A} = \frac{1}{T}\left(T - \frac{1}{4K}(\sqrt{1+8TK}-1)\right)\tilde{E} \tag{9}$$

We use *Equation 9* to fit a monomer-dimer model to the measured $E_D/x_A$ while optimizing for the two adjustable parameters: the dimerization constant $K$, and the value of structural parameter Intrinsic FRET.

The dissociation constant $K_{diss}=1/K$ is reported in units of receptors.$\mu$m$^{-2}$ in *Table 1*. The free energy of dimerization (dimer stability) $\triangle G$ is calculated from the dimerization constant $K=1/K_{diss}$. The standard state is defined as nm$^2$.receptor$^{-1}$ (*Chen et al., 2010*), and therefore:

$$\Delta G = -RT\ln\left(\frac{10^6}{K_{diss}}\right) \tag{10}$$

## Cell culture, transfection, and activation with VEGF ligands for Western blot experiments

Chinese Hamster Ovary (CHO) cells and Human Embryonic Kidney (HEK) 293T cells were cultured at 37°C with 5% $CO_2$ for 24 hr. Transfection was carried out using Fugene HD transfection reagent (Roche Applied Science), following the manufacturer's protocol. Cells were cotransfected with 0.1–1 ug of wild type or mutant DNA plasmid constructs encoding VEGFR-2.

The cells were starved in medium lacking fetal bovine serum (FBS) for 24 hr. The cells were then treated with 3000 ng.mL$^{-1}$ of human VEGF ligands. After incubating for 13 min at 37°C with ligand, the samples were placed on ice prior to lysis. The cells were treated with lysis buffer (25 mM Tris-HCl, 0.5% Triton X-100, 20 mM NaCl, 2 mM EDTA, phosphatase inhibitor and protease inhibitor, Roche Applied Science). Lysates were collected following centrifugation at 15,000 g for 15 min at $4deg$C and loaded onto 3–8%NuPAGE Novex Tris–Acetatemini gels or 4–12% NuPAGE Novex Bis-Tris Plus Gels (Invitrogen, CA). The proteins were transferred onto a nitrocellulose membrane, and blocked using 5% milk in TBS. VEGFR-2 total protein levels were assessed using antibodies against VEGFR-2 (55B11; #2479, Cell Signaling Technologies) or the HA tag (3F10, Roche Applied Science). Phosphotyrosine levels were assessed using anti-phospho-VEGFR-2 antibodies (Tyr1054/1059: D5A6; #3817 or Tyr1175: D5B11; #3770, Cell Signaling Technologies). Endogenous VEGF-A levels were detected by staining with anti-VEGF-A antibodies (PA1080, Boster Biological Technology Co). This was followed by anti-rabbit HRP conjugated antibodies (W4011, Promega) or goat anti-rat HRP conjugated antibodies (sc-2006, Santa Cruz Biotechnology). All primary antibodies were used at 1:1000 dilution and secondary antibodies at 1:2500 dilution according to the manufacturers' protocols. The anti-HA antibodies were used at 1:3000 dilution. The proteins were detected using the Amersham ECL detection system (GE Healthcare).

### Cross-linking experiments

Following 24 hr starvation, CHO cells expressing full length VEGFR-2 were subjected to cell surface cross-linking using 2 mM BS$^3$ (Pierce), a membrane impermeable crosslinker, for 30 min at room temperature. The samples were quenched in 20 mM Tris-HCl for 15 min. After one rinse with ice-cold PBS, the cells were lysed and the receptors were detected via Western blotting, using antibodies against VEGFR-2 (55B11: #2479, Cell Signaling Technologies), followed by secondary HRP conjugated anti-rabbit antibodies (W4011, Promega).

## Acknowledgements

This work was supported by grants NIH GM68619, NIH GM95930, and NSF MCB1157687 to KH We thank Drs. Feilim Mac Gabhann and Michael Edidin for many useful discussions and for reading the manuscript prior to publication. We thank Christopher King for help with data analysis. K B-H thanks the Swiss National Science Foundation (grant 31003A-130463) and Oncosuisse (grant OC2 01200-08-2007) for continuous support of his work.

# Additional information

## Funding

| Funder | Grant reference number | Author |
| --- | --- | --- |
| Schweizerischer Nationalfonds zur Förderung der Wissenschaftlichen Forschung | 31003A-130463 | Kurt Ballmer-Hofer |
| Oncosuisse | OC2 01200-08-2007 | Kurt Ballmer-Hofer |
| NIH Office of the Director | GM068619 | Kalina Hristova |
| NIH Office of the Director | GM095930 | Kalina Hristova |
| NSF Office of the Director | MCB1157687 | Kalina Hristova |

The funders had no role in study design, data collection and interpretation, or the decision to submit the work for publication.

## Author contributions

SS, Conception and design, Acquisition of data, Analysis and interpretation of data, Drafting or revising the article; KB-H, Conception and design, Drafting or revising the article; KH, Conception and design, Analysis and interpretation of data, Drafting or revising the article

## Author ORCIDs

Sarvenaz Sarabipour, http://orcid.org/0000-0001-5097-5509
Kalina Hristova, http://orcid.org/0000-0003-4274-4406

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
