## [Decision Letter]

Thank you for submitting your work entitled "VEGFR-2 conformational switch in response to ligand binding" for consideration by *eLife*. Your article has been favorably evaluated by John Kuriyan (Senior Editor) and three reviewers, one of whom, Volker Dötsch, is a member of our Board of Reviewing Editors. The other two reviewers have agreed to reveal their identities: Charles Sanders and Yechiel Shai.

The reviewers have discussed the reviews with one another and the Reviewing Editor has drafted this decision to help you prepare a revised submission.

Summary:

There are two accepted models for the activation of VEGFR-2, the "diffusion-based" in which RTKs exist as monomers in the absence of ligand. Ligand binding induces its dimerization. The second model is the "preformed dimer model", in which the receptors can form dimers in the absence of ligand but it exists in a monomer-dimer equilibrium in the absence of ligand. Ligand binding promotes structural changes in the ecto domain that trigger transmembrane signaling.

Here the authors used quantitative FRET and biochemical analysis, to show directly that the full length VEGFR-2 forms dimers also in the absence of ligand and when expressed at physiological levels. Furthermore, these dimers are also phosphorylated. In addition, ligand binding causes a change in the conformation of the trans-membrane (TM) domain, leading to increased phosphorylation. They further demonstrated that Inter-receptor contacts within the extracellular and TM domains are critical for the formation of the unliganded dimer, as well as for the transition to the ligand-bound active conformation.

All reviewers agree that the manuscript reports interesting data. However, some questions remain that should be addressed in a revised manuscript.

Essential revisions:

1) GFP variants are known to have a tendency to dimerize. Have the authors used monomeric forms of these fluorescent proteins or have taken into account this effect?

2) Some of the GFP variant proteins are not fully processed and therefore remain dark. Have the authors taken this into account?

3) What is the model of how a further separation of the C-terminal ends of the TM helices creates efficient cross phosphorylation? This seems to me rather counter intuitive. Could asymmetric kinase domain dimers play a role?

4) Both the Abstract and sections of the text (c.f. subsection “Ligand binding induces a structural change in the TM domain of the VEGFR-2 dimer”) refer to the presence of receptor phosphorylation under basal (ligand-free) conditions. What is the significance of basal phosphorylation? Does it induce basal signaling? Is it a different class (different sites?) of modification than that induced by growth factor binding?

5) The standard deviations reported in the table are mostly very small-often under 5% relative deviations (this seems very clear from the green and red data of Figure 1, for example). These values seem to reflect the SD calculated by the data fitting program but, I assume, greatly under-represents the actual experimental uncertainties of the reported measurements. This should be addressed.

6) One thing that seems to be missing in the Discussion is integration of the major conclusions of this paper with what is now known about the signaling mechanisms of other RTK growth factor receptors.

---

## [Author Response]

Essential revisions:

*1) GFP variants are known to have a tendency to dimerize. Have the authors used monomeric forms of these fluorescent proteins or have taken into account this effect?*

Yes, we use monomeric YFP and monomeric mCherry. In fact, we ourselves have introduced the A206K mutation in YFP to render it monomeric. Furthermore, we have published control experiments for ErbB truncated variants, labeled with these fluorescent donors and acceptors, that show no interactions (King et al., 2014). Thus, we know that the fluorescent proteins that we use do not induce the dimerization of the membrane proteins that they are attached to. In the revised paper, we have added “Both fluorescent proteins are monomeric, and do not drive interactions in the membrane (King et al., 2014)” to the Methods, subsection “Plasmids”, first paragraph.

*2) Some of the GFP variant proteins are not fully processed and therefore remain dark. Have the authors taken this into account?*

We have observed such effects in collaborative work with soluble fluorescent proteins. Sometimes the ratio of donor to acceptor concentrations for a linked donor-acceptor construct is not 1. However, we have seen no obvious indications of such effects for membrane proteins. We believe that the misfolded membrane proteins do not pass the quality control of the ER and do not make it to the plasma membrane and thus to our plasma membrane derived vesicles. On the other hand, misfolded soluble fluorescent proteins are made in the cytoplasm and, perhaps, are not destroyed efficiently as they are over-expressed.

*3) What is the model of how a further separation of the C-terminal ends of the TM helices creates efficient cross phosphorylation? This seems to me rather counter intuitive. Could asymmetric kinase domain dimers play a role?*

To the best of our knowledge, the VEGFR-2 results are consistent with the current model of EGFR TM domain dimer response to ligand. Upon ligand binding, the C- termini of the TM domains are believed to move away from each other (See for instance Figure 6 and Figure 7 - for “inactive dimer - in Arkhipov et al., 2013). In one model this creates “space” for the asymmetric EGFR kinase dimer to form. However, our experiments do not tell us much about the kinase domains, so we cannot speculate about a possible asymmetric kinase dimer at this point.

It is noteworthy that the movement of the VEGFR-2 TM C-termini in response to ligand, observed here, is the opposite of the one observed for FGFR TM domain C-termini with the same experimental technique. In the revised manuscript (Discussion, subsection “Ligand binding promotes reorientation of the receptors in the dimer”, fourth paragraph), we have added:

“A similar mechanism of activation, involving an increase in TM helix C-termini separation, has been proposed for EGFR (Arkhipov et al., 2013; Bocharov et al., 2008; Endres et al., 2013). For the FGF receptors, however, the ligand-induced changes involve a decrease in the separation between the TM domain C-termini (Sarabipour and Hristova, 2016). Decrease in the separation of the TM domains upon ligand binding has also been observed for the constitutively dimeric IGF-1R (Kavran et al., 2014). Thus, it appears that ligand-induced structural changes are a general feature of RTK activation, but the exact nature of the structural change varies for the different receptors.”

*4) Both the Abstract and sections of the text (c.f. subsection “Ligand binding induces a structural change in the TM domain of the VEGFR-2 dimer”) refer to the presence of receptor phosphorylation under basal (ligand-free) conditions. What is the significance of basal phosphorylation? Does it induce basal signaling? Is it a different class (different sites?) of modification than that induced by growth factor binding?*

Work in the literature has demonstrated that downstream kinases can be phosphorylated in endothelial cells in the absence of ligand. Basal phosphorylation occurs on Y1054/1059, Y1175 and Y1214. The phosphorylation of these tyrosines is increased upon VEGF treatment. In the revised manuscript (Discussion, subsection “VEGFR-2 forms dimers in the plasma membrane in the absence and presence of ligand”, first paragraph), we have added:

“The basal phosphorylation occurs on Y1054/1059 and Y1175 (Figure 6), as well as on Y1214 (Lamalice et al., 2006). Consistent with these observations, phosphorylation of downstream kinases has been reported in endothelial cells in the absence of ligand (Labrecque et al., 2003; Oubaha and Gratton, 2009).”

*5) The standard deviations reported in the table are mostly very small-often under 5% relative deviations (this seems very clear from the green and red data of Figure 1, for example). These values seem to reflect the SD calculated by the data fitting program but, I assume, greatly under-represents the actual experimental uncertainties of the reported measurements. This should be addressed.*

The reported errors are not standard deviations, they are standard *errors* determined from the MATLAB fit. This is now stated in the legend to Table 1. They are small because we have a very large number of data points (vesicles) to reduce random error. In previous work, we have analyzed the errors in our measurements, and we have shown that they are due to white noise (Chen et al., 2010). Because of this, the errors are reduced by making multiple measurements. Random errors cannot be eliminated completely, but they can be reduced by collecting a large number of data points (>300) for each experimental condition.

To reduce systematic errors, we always calibrate the microscope by imaging purified fluorescent proteins at the start of each experiment (described in the Methods section). To determine if systematic errors play a role in our measurements, at least three independent experiments for each variant are performed on different days, and measurements with multiple variants are performed on the same day. We see no systematic errors and thus the white noise is the major source of error in our experiments.

6) One thing that seems to be missing in the Discussion is integration of the major conclusions of this paper with what is now known about the signaling mechanisms of other RTK growth factor receptors.

We have expanded the Discussion to address this concern. For instance, we have added to the Discussion, subsection “VEGFR-2 forms dimers in the plasma membrane in the absence and presence of ligand”, first paragraph:

“Recent work has demonstrated that other RTKs, such as EGFR and the FGF receptors (Chung et al., 2010; Comps-Agrar et al., 2015; Lin et al., 2012; Low-Nam et al., 2011; Sarabipour and Hristova, 2016) also form unliganded dimers. A new model of RTK activation, which includes unliganded dimers of various stabilities as intermediates, has been proposed (Sarabipour and Hristova, 2016). The behavior of VEGFR-2 observed here is consistent with this new model.”

We have further added to the Discussion, subsection ligand binding promotes reorientation of the receptors in the dimer, fifth paragraph: “The VEGFR-2 case is thus quite different from the case of the FGFR dimers, which exhibit different structural and biological responses to the ligands fgf1 and fgf2 (Sarabipour and Hristova, 2016).”